# High Resolution 3-D Imaging of Mesospheric Sodium (Na) Layer Utilizing a Novel Multilayer ICCD Imager and a Na Lidar

**Xuewu Cheng [1], Guotao Yang [2], Tao Yuan [3,*], Yuan Xia [4], Yong Yang [1], Jiqin Wang [1], Kaijun Ji [1], Xin Lin [1], Lifang Du [2], Linmei Liu [1], Kaijie Ji [1] and Faquan Li [1]**

[1] Innovation Academy for Precision Measurement Science and Technology, West No 30, Xiaohongshan, Wuhan 430071, China; lidar@apm.ac.cn (X.C.); yangyong@apm.ac.cn (Y.Y.); wangjiqin18@mails.ucas.ac.cn (J.W.); jikaijun16@mails.ucas.ac.cn (K.J.); linxin@apm.ac.cn (X.L.); liulinmei@apm.ac.cn (L.L.); jikaijie12138@163.com (K.J.); lifaquan@apm.ac.cn (F.L.)

[2] National Space Science Center, Chinese Academy of Sciences, NO.1 Nanertiao, Zhongguancun, Haidian District, Beijing 100190, China; gtyang@spaceweather.ac.cn (G.Y.); lfdu@nssc.ac.cn (L.D.)

[3] Center for Atmospheric and Space Sciences, Utah State University, Logan, UT 84322-4405, USA

[4] School of Electronic Engineering, Nanjing Xiaozhuang University, Nanjing 211171, China; xiayuanxxyy@njxzc.edu.cn

* Correspondence: titus.yuan@usu.edu; Tel.: +1-(435)-797-2959

**Abstract:** Equipped with a 1-meter Cassegrain telescope with 6.2 meter focal length and an electronically gated Intensified Charge-Coupled Device (ICCD), a multilayer Na imager is designed and developed at Wuhan in China. This novel instrument has successfully achieved the first preliminary 3-D image of the mesospheric Sodium (Na) layer when running alongside a Na lidar. The vertical Na layer profile is measured by the lidar, while the horizontal structure of the layer at different altitudes is measured by the ICCD imaging with a horizontal resolution of ~3.7 urad. In this experiment, controlled by the delay and width of the ICCD gating signal, the images of the layer are taken with three-second temporal resolution for every 5 km. The results show highly variable structures in both the vertical and horizontal directions within the Na layer. Horizontal images of the Na layer at different altitudes near both the permanent layer (80–100 km) and a sporadic Na layer at 117.5 km are obtained simultaneously for the first time. The Na number density profiles measured by the lidar and those derived from this imaging technique show excellent agreement, demonstrating the success of this observational technique and the first 3-D imaging of the mesospheric Na layer.

**Keywords:** 3-D imaging; mesospheric Na layer; small-scale dynamics

## 1. Introduction

The mesospheric Sodium (Na) layer in the mesosphere and lower thermosphere (MLT) at an altitude of between ~80 km and 110 km is mainly generated through the combination of meteor ablation and multibody chemical reactions involving several Na compounds and ion species in this region [1]. Due to the high abundance and large scattering cross section of this metallic Na, the layer has become an ideal target for laser guide star systems [2] and for Na lidars measuring the temperature and wind field variations in the MLT [3,4]. Na lidar measurements over the past a few decades have significantly advanced our understanding of the dynamics and chemistry in this region [5]. Recent progress on Na lidar technology has greatly improved the sensitivity of the lidar system, and allowed unprecedented investigations to be undertaken, such as of small-scale atmospheric dynamics revealing an overturning structure in the layer due to gravity wave breaking-induced instability [6–8]. For example, Guo et al.

studied the atmospheric turbulence mixing and vertical transport mechanisms using high resolution Na lidar measurements [9].

However, as a single point observational technique, the Na lidar can conduct these valuable high resolution measurements only in the vertical direction, while important horizontal information across the layer altitude range, critical for studies of small-scale atmospheric dynamics, is inaccessible by the lidar observation alone. On the other hand, passive airglow imaging instruments can measure the horizontal variations within their field of view (FOV) around the peak altitudes of the layer of hydroxyl (OH), Na and atomic oxygen (O) in the MLT [10]. Their receiving signals, however, have to be averaged throughout the whole layer vertically due to relatively weak airglow signals, losing critical information regarding vertical variations. Many attempts to overcome this restraint, such as looking at different nightglow layers and tomographic reconstruction, have proven to be difficult and are limited to a few cases [11–13]. In this paper, taking advantage of the much stronger laser-induced fluorescent signals from Na atoms within the MLT, we demonstrate a new technique, the Multilayer Na imager, which has the ability to image the horizontal structure of the layer at different altitudes, achieving the first 3-D imaging of the mesospheric Na layer. The system, running alongside a high power Na lidar, consists of a large aperture telescope and an electronically gated, high sensitivity Intensified Charge-Coupled Device (ICCD). This new observation capability further advances our understanding of some of the most fundamental small-scale atmospheric processes. The instruments involved in this investigation are described in Section 2, followed by the new results obtained in the campaign summer 2017 in Section 3. These results are discussed in detail in Section 4. Section 5 presents the summary of this new investigation.

## 2. Instruments and Methods

The multilayer Na imager needs to operate alongside a Na lidar to image the laser-induced fluorescence signals from the Na layer at different altitudes. The lidar laser pulses "illuminate" the atomic Na within the layer through laser-induced fluorescence processes, and the imager can take pictures of the horizontal structure of the layer section at different altitudes "illuminated" by the laser pulses. Figure 1 shows a diagram of the two systems running simultaneously. The critical parameters of the lidar system and the Multilayer Na imager are listed in Table 1.

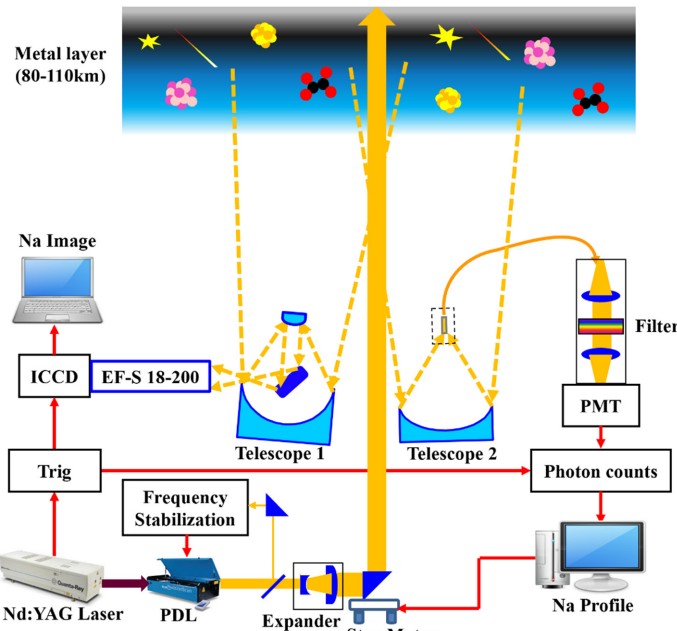

**Figure 1.** The system layout of the Multilayer Na imager: the Na imager (left of the orange lidar beam) alongside of a high power Na lidar (right of the orange lidar beam).

**Table 1.** Main technical specifications of the Na imager and the Na lidar.

| | Name | Imaging Detection | Profile Detection |
|---|---|---|---|
| Launch | Wavelength | 589.158 nm | |
| | Line width | 1.2 pm | |
| | Energy | ~40 mJ | |
| | Beam Divergence | ~0.2 × 0.6 mrad | |
| | Pulse Width | 8~10 ns | |
| | Repetition Rate | 30 Hz | |
| Receiver | Telescope | Cassegrain | Newtonian |
| | Aperture | Φ1 m | Φ1 m |
| | Focal length | 6.2 m | 2.1 m |
| | Field of view | ~3.8 mrad | ~0.7 mrad |
| | Emitter − Receiver distance | 20 m | 6 m |
| Detection | Filter bandwidth | - | 1 nm |
| | Effective detection area | 13 × 13 mm | Φ5 mm |
| | CCD image pixels | 1024 × 1024 | - |
| | Quantum efficiency | ~45% | ~40% |

## 2.1. The Na Lidar

The Na lidar was developed and has been operating at Wuhan Institute of Physics and Mathematics in Wuhan, China (30.6 °N, 114 °E). The lidar transmitter is a pulsed dye laser (Sirah Cobra-Stretch by Sirah Lasertechnik in Grevenbroich, Germany) pumped by a 30-Hz Nd:YAG laser (Spectra-Physics Pro 290-30 by Spectra-Physics, Lnc. in Mountain View, CA, USA), emitting laser pulses at 589 nm, the resonate wavelength of the Na D2 line. The energy per laser pulse is ~40 mJ with a laser linewidth of ~1.2 pm and a pulse lifetime of less than 7 ns, i.e., considerably less than the Na lifetime in its 3P state (16 ns). The wavelength of the laser pulse is controlled by sending ~1% of the 589 nm laser power into a temperature controlled Na absorption cell to monitor the frequency in real time. This setup forms a feedback loop by adjusting the angle of the grating inside the pulse dye laser and, thus, locking the laser frequency at the Na $D_2$ line (589.158 nm). Due to the rectangular shape of the dye cell in the last stage of the Sirah dye laser, the output beam spot is elliptical in shape. To limit the divergence of the output laser beam and increase the lidar receiving efficiency (without saturating the atomic Na in the mesospheric Na layer), a five-time beam expander is placed after the dye laser, leading to a ~0.2 mrad divergence angle in the short axis ($\grave{E}_{short}$) and a ~0.6 mrad divergence angle in the long axis ($\grave{E}_{long}$). The laser generates laser-induced fluorescence signals from the Na atoms in MLT and "illuminates" the mesospheric Na layer. The lidar receiver collecting these fluorescence signals is a one-meter diameter Newtonian telescope with a focal length of 2.1 m, placed ~6 meters away from the outgoing Na lidar laser beam. This off-axis setup can considerably decrease the scattering signals from the lower atmosphere, which would easily saturate the lidar detector. A 1.5-mm diameter, multimode filter is placed at the focal plane of the Newtonian telescope to collect the Na lidar signals. This defines the lidar receiver FOV as ~0.71 mrad, i.e., slightly larger than the divergence angle of the long axis of the laser beam. This lidar FOV design makes it highly efficient, receiving the lidar signals while minimizing the sky background. The fiber then guides the lidar signals to a photo multiplier tube (H7421-40 by Hamamatsu Photonics K.K. in Hamamatsu City, Japan) that converts the photon signals to electronic pulses. These pulses are then counted by a fast digital counting card (MCS-PCI by AMETEK Inc. in Berwyn, PA, USA) to generate the photon counting profiles saved in the data taking computer.

### 2.2. The Multilayer Na Imager

Because the laser "illuminated" section of the mesospheric Na layer occupies only a small horizontal part of the layer, the high resolution imaging of this section requires a high quality lens system with high-magnification zoom capability and a telescope with long focal length (FOV control to minimize the noise outside of this "illuminating" section). Thus, a 1-meter diameter parabolic curved Cassegrain telescope (Nanjing Institute of Astronomical Optics & Technology, National Academy of Science, Nanjing, China) with a focal length of ~6.2 m is chosen and installed ~20 m from the lidar outgoing laser beam. This off-axis setup allows images of the Na layer at different altitudes to form at different locations on the ICCD detection area, which will be discussed later. A Canon EF-S 18-200 lens is placed right after the field stop to image the telescope focal plane onto the high sensitivity ICCD (EMICCD, model PI-MAX4-1024EMB, by Princeton Instruments, Inc. in Trenton NJ, USA) that has an effective detection area of 13 × 13 mm and 1024 × 1024 pixels. Triggered by the lidar, the imaging at different altitudes is achieved by precisely adjusting the gate delay time (relative to the Na lidar trigger signal), while the vertical resolution can be set by the gate width (round trip photo traveling time within the vertical resolution). Because the photons are laser-induced fluorescence, the gate delay time needs to be set as the photon arriving time. For example, photons coming from 100 km altitude have a round trip traveling time of 667 μs. The data collecting software is provided by Princeton Instruments, and images are saved in the .tif (.fit) format. As shown in Figure 1, the ICCD is triggered by the Q-Switch signal of the Nd:YAG laser. Once the imaging altitude is chosen (delay time of the electronic gate), the ICCD exposure time, decided by gate width, can be set to decide the vertical resolution. For example, the ~ 33¦Ìs exposure time determines a vertical resolution of 5 km (round-trip traveling time of photons within this range). Because the signal to noise ratio (S/N) is low through single exposure, we set the ICCD to keep taking pictures for 3 s (~100 laser pulses), and then integrate these pictures into one data file, leading to a vertical resolution of 5 km and temporal resolution of 3 s. The system then turns to another altitude and repeats this process. The ICCD software package is provided by Princeton Instruments Inc., LightField Software, which can be integrated into National Instruments' LabVIEW interface. Overall, the instrument can capture images from an altitude range between 20 km and 120 km (the upper limit depends on the upward Na layer extension). The images below ~50 km are generated by Rayleigh scattering, while those between ~80 km and 120 km are the laser-induced fluorescence signals. It is worth noting that between ~50 km and 80 km, the dramatically decreasing atmospheric density makes the Rayleigh scattering too weak for the ICCD to generate any image with a decent signal-to-noise ratio.

To project the image on the telescope focal plane onto the ICCD active area, a Canon zoom length with 200 zoom capability is adapted here to match the telescope FOV. By utilizing its maximum zoom capability, we projected a Thorlabs steel ruler that is placed on the telescope's focal plane onto the ICCD active area (Figure 2) to quantify the resolution of this Na imager. The two dots are parts of a symbol on the ruler. As the figure shows, the ICCD covers an area of 23.5 mm × 23.5 mm in the telescope's focal plane, corresponding to 3.8 mrad FOV. Based on this, considering that the ICCD active area has 1024 × 1024 pixels (13 mm × 13 mm), the instrumental horizontal resolution is ~3.7 μrad per pixel. However, it is worth noting that the astronomical measurements above an astronomy observatory are mostly about 4.8 μrad, i.e., larger than this instrumental resolution. Since the Cassegrain telescope used for this experiment does not have an adaptive optics system, the actual horizontal resolution is worse than the calculated results above.

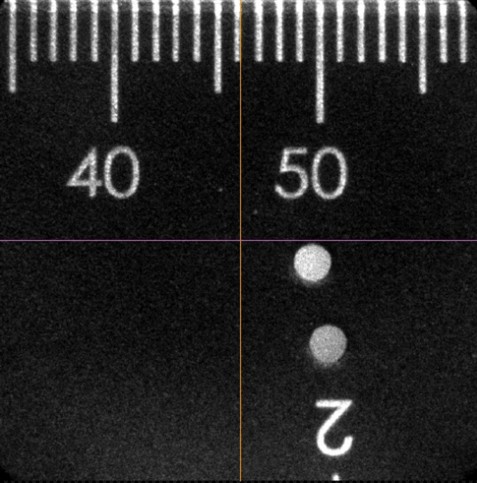

**Figure 2.** The image of a Thorlabs steel ruler on the ICCD active area by the lens system of the Na imager. One tick mark represents 1 mm.

The basic optical principle of the Na imager is illustrated in Figure 3, where $L$ is the object distance and $l'$ is the image distance, $H$ is height of object and $h'$ represents height of the image. The focal length of the telescope is 6.2 m. Based on the Gaussian lens formula of geometric optics under paraxial approximation, $\frac{1}{l'} + \frac{1}{L} = \frac{1}{F}$, the image distance, $l'$, for the Rayleigh signal at 30 km altitude and Na fluorescent signal at 100 km altitude are 6201.3 mm and 6200.4 mm, respectively, leading to a distance of 0.9 mm between the two images, i.e., much less than the focal length, $F$. Thus, to simplify, we can treat $l' = F = 6.2$ m . Because of the laser beam divergence, the sizes of the beam spots (section of Na layer "illuminated" by laser) at different altitudes are different. Since the dimension of the beam spot at far field, for example, at an altitude of 100 km, is much larger than that at near field (right out of the beam expander the beam spot diameter along the short axis is $H_0 = 0.05$ m), we can ignore the contribution from the dimension of the near field, and calculate the size of the image of the far field beam spot on the telescope focal plane as:

$$h' = \frac{H \cdot l'}{L} = \frac{(H_0 + \theta \cdot L) \cdot l'}{L} \approx \theta_{short} \cdot l' = 1.24 \ mm \ (\text{along the short axis}), \tag{1}$$

for the direction along the short axis. Therefore, the size of the layer image on the Cassegrain telescope focal plane becomes almost independent of the altitude, and is mostly decided by the product of the laser divergence angle and the telescope focal length. The same applies to the far field dimension along the long axis; the length of the far field beam spot along the long axis is also a constant, 3.72 mm, on the telescope focal plane.

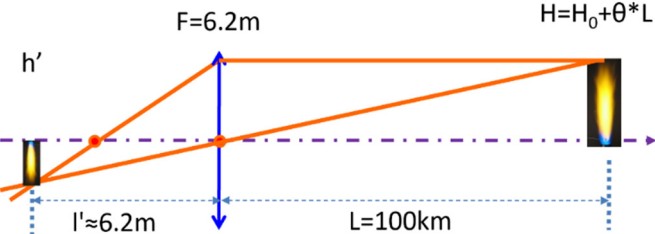

**Figure 3.** Geometric Optics diagram of the imaging the Na layer using a Cassegrain telescope with 6.2 m focal length.

As we mentioned earlier, since the Cassegrain telescope for the horizontal layer observation is ~ 20 m away from the lidar laser beam, the laser beam spots at different altitudes form at different locations on the ICCD active area. Our calculation shows that the distances between the center locations

of the beam spots at altitudes of 20 km, 80 km, 100 km and 120 km are 112 pixels, 14 pixels and 9 pixels, respectively, as shown in Table 2. During the data analysis process, when comparing different images, this information can be easily accounted for.

**Table 2.** Image deviation on the ICCD active area at different altitudes.

| Altitude (km) | 20 | 80 | 100 | 120 |
|---|---|---|---|---|
| Image deviation to the optical axis (pixel) | 270 | 68 | 54 | 45 |
| Relative deviation (pixel) | | 212 | 14 | 9 |

## 3. Results

### 3.1. Horizontal Variations of the Na Layer at Different Altitudes Across the Layer

Figure 4 demonstrates the multilayer imaging of the mesospheric Na layer conducted at two different local times on the night of 7 June, 2017. The figure shows 10 images of the Na layer illuminated by the laser from 77.5 km to 122.5 km every 5 km for each local time. It should be pointed out that these altitudes reflect the center altitude of the 5-km window. The climatological Na number density profile in the MLT is close to a Gaussian shape, peaking near 90 km altitude [14]. However, short-term variability can lead to highly variable Na number density variation across the layer altitudes. Since the laser-induced Na fluorescence is directly proportional to Na number density, images near the peak of the layer, i.e., at 87.5 km and 92.5 km around the peak altitude of the layer, have the strongest intensities and the best S/N. The near elliptical shape reflects the laser beam power distribution across the layer, because the laser beam profile coming out of the dye laser is almost elliptical (see Section 2.1). A secondary sporadic Na layer between 107.5 km and 117.5 km can be clearly seen in both plots, although the intensity is much weaker than that at 92.5 km due to the lower Na number density. Such secondary high altitude metal layers have been observed and reported around the globe by several lidar systems [15–19], and provide great opportunities to study the dynamic coupling processes between the neutral atmosphere and the ionosphere. The S/N across the image is not uniform due to laser intensity distribution. Thus, the uncertainty also varies across the image. The factional uncertainties of the image, i.e., the inverse of S/N, are mostly below 10 % within the permanent Na layer, but can be as large as 50% near 100 km and near the edge of the laser spot. The uncertainties of the secondary Na layer varies considerably, depending on the Na number density.

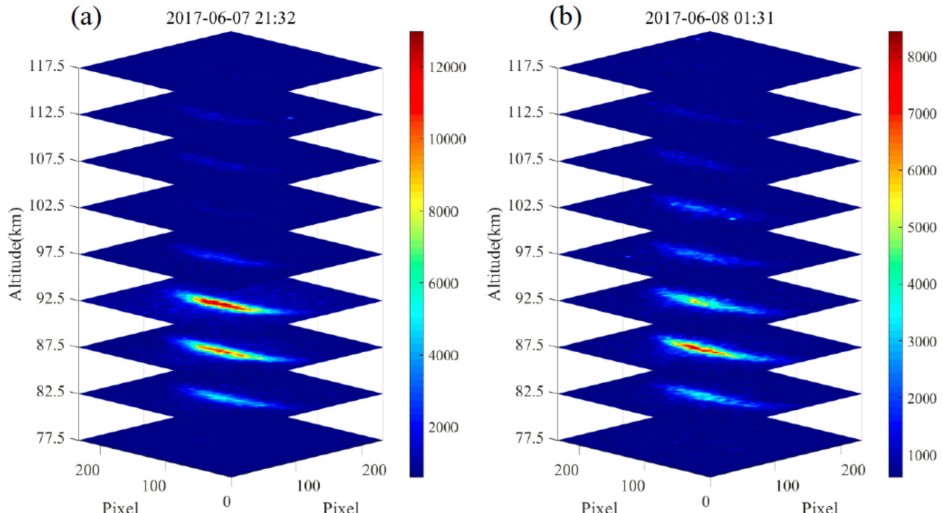

**Figure 4.** Multi-Na layer imaging experiment conducted at local time 21:32 on 7 June (**a**) and at 01:31 on 8 June (**b**) 2017, respectively. Horizontal axis is the number of pixels.

Figure 5 demonstrates simultaneous images at 20 km (Rayleigh) which were averaged through the 80–120 km altitude range. Due to the off-axis setup, these two images from different altitude ranges formed at different locations on the ICCD active area. The solid orange line represent the signal intensity distribution across the image. Each point in the lidar is the sum of all the intensities of the pixels across the horizontal line. The shape of the laser beam spot (laser power distribution) can be clearly shown in the Rayleigh signal image, which is close to a Gaussian distribution along both axes. The zoomed-in image of a star in the FOV is also shown in the plot to the right, which occupies $5 \times 6$ pixels, corresponding to 15 urad. Since most of the atmospheric distortions during the imaging process are generated in the troposphere and lower stratosphere, the Rayleigh image of the laser spot, such as that at 30 km altitude, could serve as the normalization factor, which would remove the effect on the Na images due to nonuniform laser power distribution. However, the current S/N of the Rayleigh image is still relatively low due to background noise, especially in the area outside the full-width-half-maximum (FWHM) region. To specifically measure the horizontal Na layer disturbances due to atmospheric dynamics, such as turbulence, it would be ideal to have an advanced adaptive optics system [20,21] added to this Na layer imager to account for imaging distortion due to troposphere variations. Increasing the lidar laser power, and thereby enhancing the Na fluorescence signals, could increase the S/N of the Na imaging as well, because both the laser-induced fluorescence intensity and the Rayleigh scattering intensity are linearly proportional to the laser power while the sky background is almost constant. With the current resource and instrument limitation, to confidently reveal the small-scale dynamic structure at each altitude, further data processing techniques (3-D image smoothing) and noise reduction algorithms are needed, including for the sharp noise induced by stars moving within the FOV of the imager during observations.

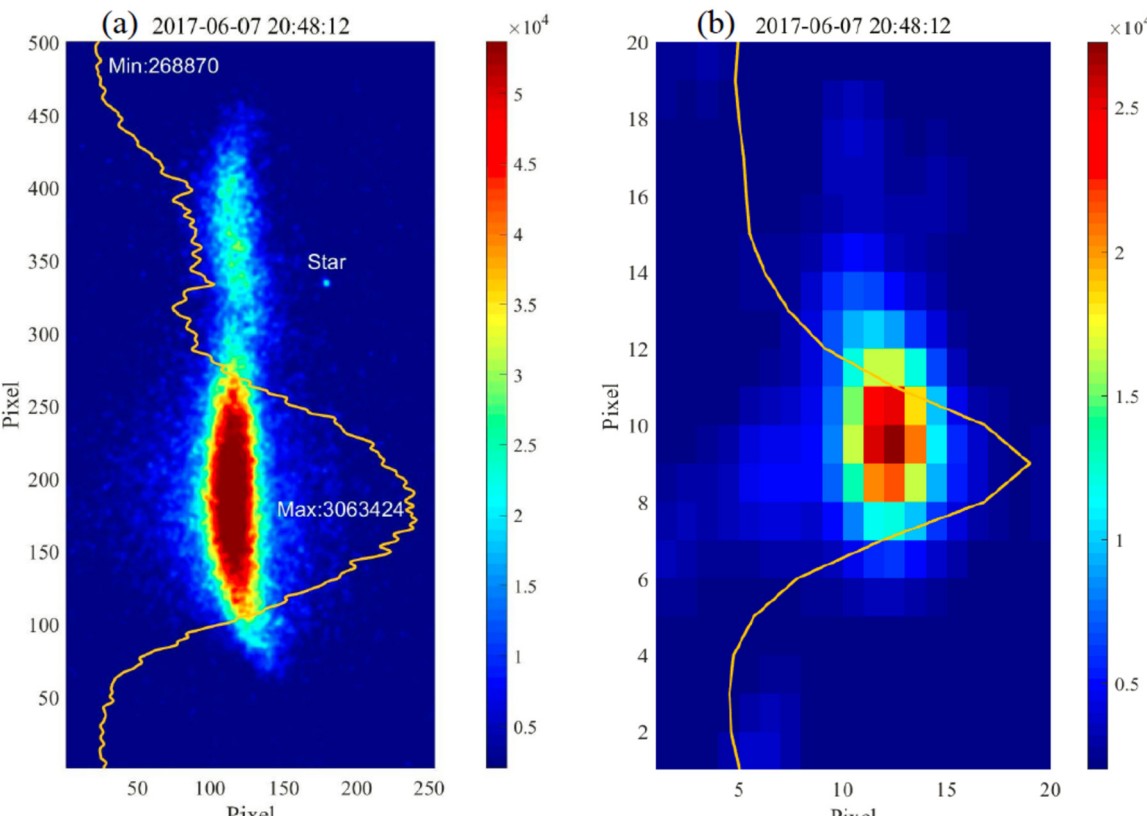

**Figure 5.** The simultaneous Rayleigh spot around 20 km altitude (80–300 pixel vertical) and Na spot (320–425 pixel vertical) averaged between altitudes of ~80 km and 120 km. The unit of both axes is "number of pixel". The image to the right is a zoomed-in picture of a star that happened to be in the FOV during the experiment.

### 3.2. Na Density Profile

In addition to the ability to reveal the horizontal structure of the layer, these Na layer imaging data, including the Rayleigh imaging, can also be utilized to calculate the Na density vertical distribution through traditional Rayleigh power normalization, because the brightness of the image (intensity of laser-induced Na fluorescence) is linearly proportional to the Na number density at each altitude, as we described earlier. A similar technique has also been utilized to derive the Na density profile from lidar observations using a continuous high power laser system [22,23]. The potential capability of the multilayer Na imager is demonstrated in Figure 6, where the Na fluorescence intensity measurements from the Multilayer Na imager at two different local time are compared with the photon counting profiles from simultaneous Na lidar observations, showing good agreement between the two instruments. It has to be pointed out that the imager intensity is scaled by a factor of $3.6 \times 10^{-5}$ after the background subtraction, to match the order of magnitude of the lidar photon profile. Thus, when running alongside of high power adaptive optics system operating at Na $D_2$ line hosted in a large astronomical observatory, this Na imager could simultaneously measure the Na number density profile, providing important information regarding the Na beacon altitude [24].

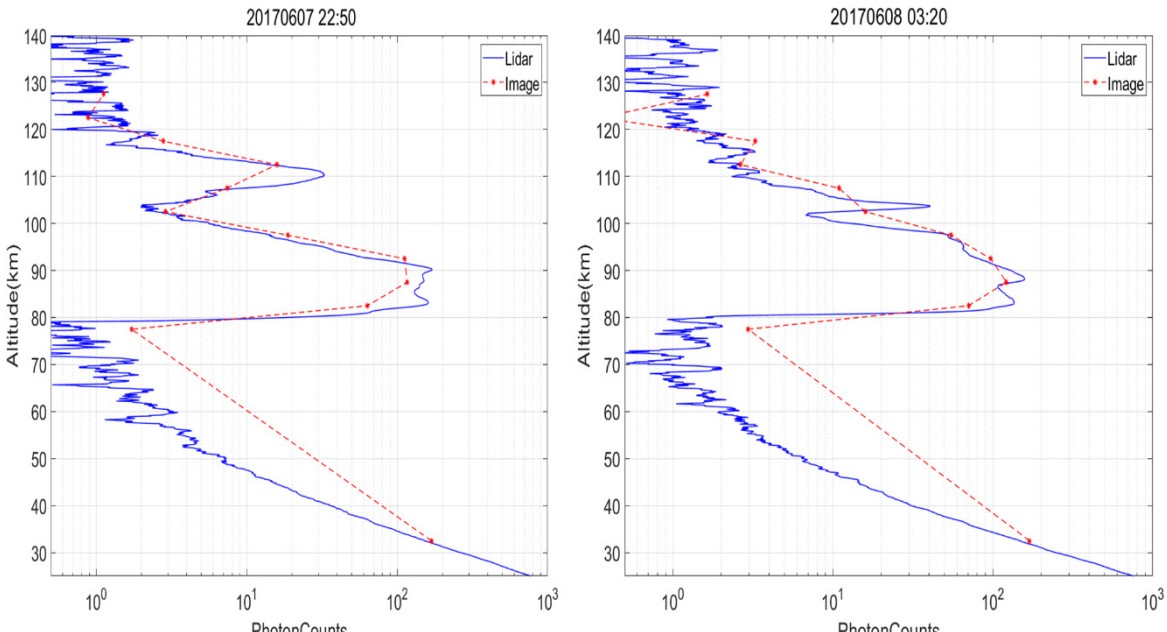

**Figure 6.** The photon counting profiles derived from the Na lidar measurements (blue solid-line) and the Na fluorescence intensity profiles measured simultaneously by the collocated multilayer Na imager (red dashed-line).

## 4. Discussion

Imaging of the mesospheric Na nightglow, averaged over the whole Na layer, has been conducted over the past several decades, revealing various small-scale dynamic structures [7]. Compared with standard nightglow imagers with a typical horizontal resolutions of a few km and no vertical profile capability, this new instrument is capable of imaging horizontal variations in the Na layer at different altitudes with resolutions of tens of meters and temporal resolutions of several seconds, achieved near the layer peak altitude. Utilizing this unique multilayer technique, the imaging of the horizontal structure of the secondary Na layer in the lower thermosphere between 100 km and 120 km becomes possible and could potentially provide an unprecedented opportunity to study the much needed small-scale atmospheric waves and the associated neutral dynamic features in this altitude range, which was previously impossible to achieve. A secondary Na layer was first reported by Gong et al. [25] in 2003, mostly between ~105 km and 120 km. This prominent dynamic feature has been frequently

observed by the Na lidars around the globe in the lower thermosphere [26,27]. Recent studies have shown that the secondary Na layer appears more frequently in summer months and seems to have quite high Na density during early evening [28,29]. Although the mechanism of its formation is still unclear [30–32], it provides a new opportunity to study important ion-neutral coupling processes in the upper atmosphere. Figure 7 demonstrates the Na density variations measured by the lidar throughout the night of 7 June, 2017, which indicates a typical secondary Na layer near 110 km lasting for most of the night. It had high Na number density during the early evening hours, i.e., between ~20:00 LT and 21:00 LT, and then faded into the background in the early morning, when some dramatic disturbance occurred near 100 km after ~01:30 LT and became stronger near 03:00 LT, similar to turbulence structures due to atmospheric instabilities. Our new multilayer imaging technique provides the first picture of the horizontal structure of these intriguing features.

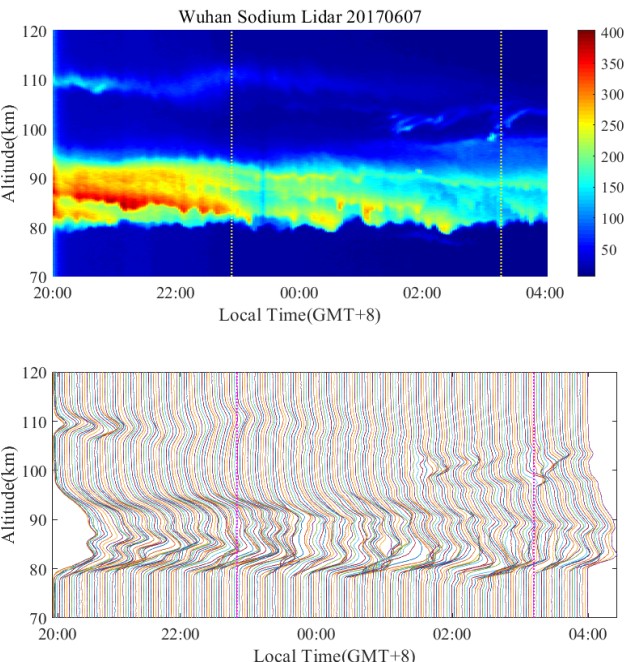

**Figure 7.** Simultaneous Na density measurements by the co-located Na lidar on 7 June, 2017. The two vertical lines in each plot mark the local time of the Na layer images in Figure 4.

One of the most important discoveries from these preliminary observations is that the horizontal structure of the secondary Na layer appears to be much more dramatic than that in the permanent mesospheric Na layer between ~80 and 95 km. As we mentioned earlier, the Rayleigh image of the laser spot (Figure 5) could be treated as the laser horizontal power distribution before entering into the Na layer, and can be utilized to calculate the absolute magnitude of the horizontal Na density perturbations in the future. Figure 8 shows the layout of the images of the Na layer horizontal structure covered by the laser spot at different altitudes. The tiny bright spots in the images of 100–105 km and 115–120 km are due to stars that happened to move into the FOV of the Na imager during the observation, and thus, should not be counted as Na signals. Compared with those in the lower altitudes, the images of 105–110 km and 110–115 km show much stronger horizontal disturbances. Keep in mind that the laser beam spot near 100 km is about 60 meters along its long axis. Therefore, without any power normalization, these pictures reveal potential horizontal modulation with a scale of ~20 m within the secondary Na layer that does not appear in either the Rayleigh image in Figure 5 or that at 30 km altitude (not shown), indicating that these horizontal irregularities are most likely due to the small-scale dynamics in the upper atmosphere. Because this paper is focusing on reporting the instrumental design and demonstrating some preliminary observations, this atmospheric dynamic topic is beyond the scope of current investigation. Furthermore, by setting up a few more identical Na layer imaging

systems surrounding the Na lidar, the signal level of the images at different altitudes within the layer could be significantly improved, providing more robust data for this field of atmospheric science.

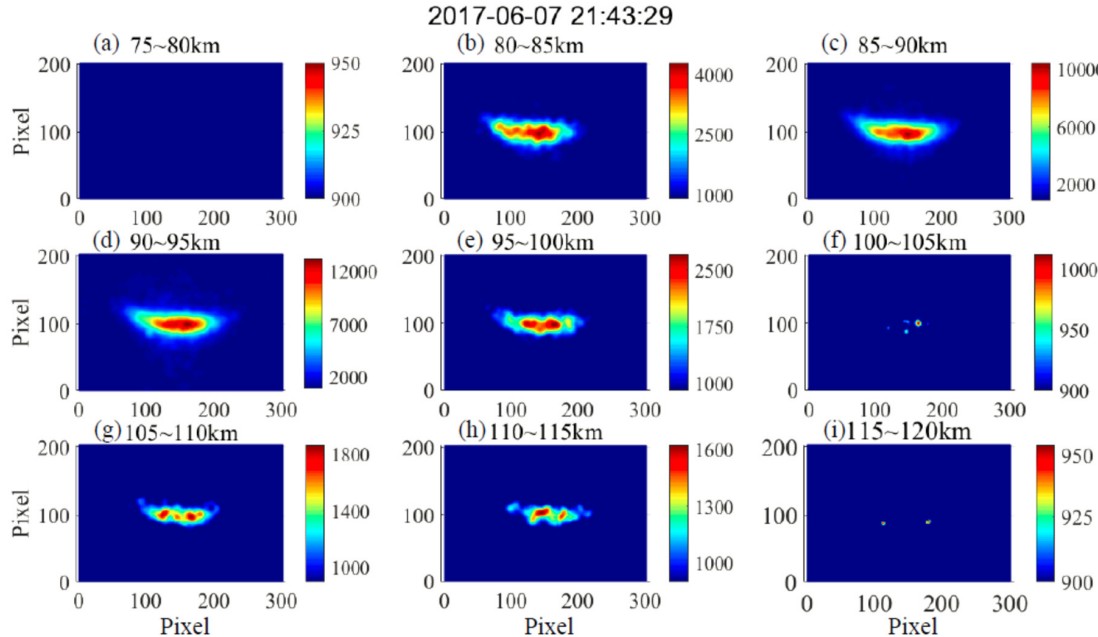

**Figure 8.** Layout of the Na layer images at six altitude ranges (every 5 km): 75–80 km (**a**), 80–85 km (**b**), 85–90 km (**c**), 90–95 km (**d**), 95–100 km (**e**), 100–105 km (**f**), 105–110 km (**g**), 110–115 km (**h**) and 115–120 km (**i**), conducted on 7 June 2017.

## 5. Conclusions

In this paper, we present a new instrument, a Multilayer Na imager operating alongside a Na lidar, which can contribute to various upper atmospheric science investigations. This technology takes advantage of strong laser-induced fluorescence signals from the Na atoms within the mesospheric Na layer in the MLT region, when they are excited by laser pulses from a high power Na lidar system. This novel Na imager provides 3-D high resolution images of the mesospheric Na layer that can reveal horizontal dynamic variation as small as ~20 m or less within the permanent layer (~80 km to 100 km). The key components include a 1-meter Cassegrain telescope with 6.2 m focal length and a highly sensitive electronic gated ICCD, which is triggered by the Nd:YAG laser of the Na lidar. The flexibility of the delay time and width of the ICCD gating signal adjustment allows the imager to take pictures of the horizontal structure of the Na layer at different altitudes with specific vertical resolution. Currently, with ~40 mJ per laser pulse, each image has a vertical resolution of 5 km (controlled by the gate width) and requires ~100 laser pulses (~3 s) to gain obtain a good S/N ration. Note that these resolutions can be modified for different scientific investigations.

During the initial observation, this Na imager system took pictures of the mesospheric Na layer at different altitudes within the layer. It successfully revealed the horizontal structure of the secondary Na layer near 110 km for the first time, showing different characteristics compared to those in the permanent layer below 100 km. The success of this Na layer imaging system shows great potential for its future contributions to various important upper atmosphere scientific topics, such as small-scale dynamics, turbulence mixing mechanisms, disturbances induced by meteor injection, etc.

**Author Contributions:** Methodology, X.C.; software and optical mechanical design, Y.Y.; validation, X.C.; J.W., X.L., L.D., L.L.; formal analysis, G.Y., T.Y.; Space and atmospheric science investigation, Y.X., X.C., G.Y. and T.Y.; resources, T.Y. and G.Y.; F.L., G.Y.; data curation, X.C.; writing—original draft preparation, T.Y.; writing—review and editing, T.Y.; visualization, X.C., T.Y.; supervision, X.C., G.Y., F.L.; project administration, F.L., G.Y.; funding acquisition, F.L., G.Y. and T.Y.; instrument construction: K.J. (Kaijie Ji); instrument construction and testing: K.J. (Kaijun Ji). All authors have read and agreed to the published version of the manuscript.

**Funding:** This work was funded by the National Science Foundation of China (NSFC) (Grant number 41604130, 41627804, 41827801]; Key Technologies Research and Development Program [2016YFC1400300], Key technical personnel of the Chinese Academy of Sciences.

**Acknowledgments:** The authors want to thank the Wuhan Institute of Physics and Mathematics Chinese Academy of Science for providing the infrastructure and resources to conduct this experiment.

**Conflicts of Interest:** The authors declare no conflict of interest.

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
