# Peer review of "High Resolution 3-D Imaging of Mesospheric Sodium (Na) Layer Utilizing a Novel Multilayer ICCD Imager and a Na Lidar"

_remotesensing, doi:10.3390/rs12223678_

Round 1

Reviewer 1 Report

Dear authors, 
thank you very much for the original and very interesting work. The approach of 3D imaging is novel and might be really useful for future studies of the upper atmosphere dynamics. 

I have only several minor comments on the content:  

  • Section 1: Introduction - should contain an overview of what is the purpose of the following sections
  • Section 2: Instruments and methods - should contain at least a short description of used SoftWare. How the observations are controlled and how data are managed.
  • Section 3: Results - please add a discussion about the uncertainties of the measurements.
  • Figures 4, 5, 7, 8 require the scale of used colors in appropriate units- what is the resolution of CCD? 1024 x 1024 pixels is it is specified in text or ~ 200 x ~ 300 pixels as it is displayed in Figures?
  • Figure 5 is unclear. The orange curve is not explained. Maybe it will be helpful to change the orientation of this Figure from horizontal to vertical. As of now, the x-axis represents the altitude.
  • line 265: sport -> spot 
  • please add a short status of tropospheric conditions during the reported measurements and how they were controlled
  • Figure 6 -  the numbers after the date are not clear? What is the meaning of '6' and '14'? Also, the profiles above ~25 km are displayed not above ~80 km as it is mentioned in the caption. Please, clarify also the relation between photon counts and density. The density profiles are mentioned in the caption but photon counts are displayed in the x-axis label.

Looking forward to seeing an improved manuscript.

With Kind Regards

Author Response

Dear authors, 
thank you very much for the original and very interesting work. The approach of 3D imaging is novel and might be really useful for future studies of the upper atmosphere dynamics. 

I have only several minor comments on the content:  

  • Section 1: Introduction - should contain an overview of what is the purpose of the following sections
  • Answer: We have added the description on the layout of the rest of the paper at the end of the introduction. Please see line 63-66 in the revision.
  • Section 2: Instruments and methods - should contain at least a short description of used SoftWare. How the observations are controlled and how data are managed.
  • Answer: We have added the description of the ICCD software in section 2.2. Please check line 132-134. The ICCD software package is provided by Princeton Instrument, LightField Software, which can be integrated into National Instruments’ LabVIEW interface. As we described in the paper, the ICCD is triggered by the Q-Switch signal of the Nd:YAG laser of the Na lidar. Once the imaging altitude is chosen (delay time of the electronic gate), the ICCD exposure time, decided by gate width, can be set to decide the vertical resolution. For example, the ~ 33ms exposure time decides the vertical resolution of 5 km (round-trip traveling time of photons within this range). Because the signal to noise ratio (S/N) is low through single exposure, we set the ICCD to keep taking pictures for 3 seconds (~ 100 laser pulses) and, then, integrate these pictures into one data file, leading to vertical resolution of 5 km and temporal resolution of 3 seconds, respectively. The system then turns to another altitude and repeats this process. 
  • Section 3: Results - please add a discussion about the uncertainties of the measurements.
  • Answers: Great suggestion! This discussion is placed near the end of the first paragraph in section 3.1, line 200-203 in the revision. Basically, the S/N across the image is not uniform due to laser intensity distribution. Thus, the uncertainty also varies across the image. The factional uncertainties of the image, which is the inverse of S/N, are mostly below 10 % within the permanent Na layer, but can be as large as 50% near 100 km and near the edge of the laser spot. The uncertainties of the secondary Na layer varies considerably depending on the Na number density.
  • Figures 4, 5, 7, 8 require the scale of used colors in appropriate units- what is the resolution of CCD? 1024 x 1024 pixels is it is specified in text or ~ 200 x ~ 300 pixels as it is displayed in Figures?
  • Answer: We have modified these figures based on the suggestion. Most of the Na layer/Rayleigh images only occupy one section of the ICCD, which is ~ 200 x ~ 300 pixels as it is displayed in Figures. Thus, only this section of the ICCD are shown. However, we want to point out that with different Na lidar systems that have larger laser divergence angle than current system, the occupied ICCD area would be different. For future investigations focusing on small-scale dynamics, slightly larger laser divergence angle (larger “illuminated” area in the Na layer) is actually preferred.
  • Figure 5 is unclear. The orange curve is not explained. Maybe it will be helpful to change the orientation of this Figure from horizontal to vertical. As of now, the x-axis represents the altitude.
  • Answer: We improved the description of this orange profile in the revision. It is basically the sum of the intensity of all pixels across each horizontal line (vertical line in the original manuscript). Please see line 206-208 in the revision.
  • line 265: sport -> spot 
  • Answer: Corrected. See line 275 in the revision.
  • please add a short status of tropospheric conditions during the reported measurements and how they were controlled.
  • Answer: Not sure we understand this question completely. Same as the sky condition for lidar operation, this imager needs clear sky to take these images. Since the observatory is located in metro area, the sky condition is not idea, even it is clear. In the future campaigns, we are planning to deploy the instrument to the Na lidar observatories at remote sites.
  • Figure 6 -  the numbers after the date are not clear? What is the meaning of '6' and '14'? Also, the profiles above ~25 km are displayed not above ~80 km as it is mentioned in the caption. Please, clarify also the relation between photon counts and density. The density profiles are mentioned in the caption but photon counts are displayed in the x-axis label.
  • Answer: Sorry for the confusion. The number represents the number of file recorded. In the new figures, we have changed it to specific time. These two are photon counting profiles from the lidar and corresponding imager intensities at different altitudes. We are still working on the data analysis algorithm to convert the ICCD intensity to Na number density at the moment. So we are just presenting the intensity profile in this paper. Deriving Na number density from Na lidar photon counting profile is quite straight forward. Thus, if we can establish the relation between the ICCD intensity to the lidar photon counting profile, then will be able to derive the Na number density from the ICCD measurements. As we discussed in the paper, we have to reduce the ICCD intensity by a factor of 0.000036 after the background subtraction. This factor has been quite robust for this campaign, so it could be the key factor to convert ICCD intensity to Na number density. But we need more observations to confirm it. We also clarify the description in the figure caption and manuscript. Please see the new figure caption and line 236-242 in the revision.

Looking forward to seeing an improved manuscript.

With Kind Regards

Reviewer 2 Report

finally, someone managed to build such an instrument
